# The Utility of Urinary Titin to Diagnose and Predict the Prognosis of Acute Myocardial Infarction

**DOI:** 10.3390/ijms25010573

**Published:** 2024-01-01

**Authors:** Miharu Arase, Nobuto Nakanishi, Rie Tsutsumi, Ayuka Kawakami, Yuta Arai, Hiroshi Sakaue, Jun Oto

**Affiliations:** 1Department of Emergency and Critical Care Medicine, Graduate School of Biomedical Sciences, Tokushima University, Tokushima 770-8503, Japanyutaarai@hotmail.com (Y.A.); joto@tokushima-u.ac.jp (J.O.); 2Department of Disaster and Emergency Medicine, Graduate School of Medicine, Kobe University, Kobe 650-0017, Japan; 3Department of Nutrition and Metabolism, Graduate School of Biomedical Sciences, Tokushima University, Tokushima 770-8503, Japan

**Keywords:** acute myocardial infarction, urinary titin, muscle

## Abstract

Early detection and management are crucial for better prognosis in acute myocardial infarction (AMI). Serum titin, a component of the sarcomere in cardiac and skeletal muscle, was associated with AMI. Thus, we hypothesized that urinary N-fragment titin may be a biomarker for its diagnosis and prognosis. Between January 2021 and November 2021, we prospectively enrolled 83 patients with suspected AMI. Their urinary N-fragment titin, serum high-sensitivity troponin I (hsTnI), creatine kinase (CK), and creatine kinase-MB (CK-MB) were measured on admission. Then, urinary titin was assessed as diagnostic and prognostic biomarker in AMI. Among 83 enrolled patients, 51 patients were diagnosed as AMI. In AMI patients who were admitted as early as 3 h or longer after symptom onset, their urinary titin levels were significantly higher than non-AMI patients who are also admitted 3 h or longer after symptom onset (12.76 [IQR 5.87–16.68] pmol/mgCr (creatinine) and 5.13 [IQR 3.93–11.25] pmol/mgCr, *p* = 0.045, respectively). Moreover, the urinary titin levels in patients who died during hospitalization were incredibly higher than in those who were discharged (15.90 [IQR 13.46–22.61] pmol/mgCr and 4.90 [IQR 3.55–11.95] pmol/mgCr, *p* = 0.023). Urinary N-fragment titin can be used as non-invasive early diagnostic biomarker in AMI. Furthermore, it associates with hospital discharge disposition, providing prognostic utility.

## 1. Introduction

Acute myocardial infarction (AMI) is caused by sudden cardiac ischemia, resulting in myocardial necrosis [1]. It is a leading cause of death and disability in the world, and is well known that in-hospital mortality increases as the time from hospital arrival to first balloon inflation (door-to-balloon time) increases [2,3]. Thus, it is very important to identify high-risk patients and diagnose AMI as early as possible. Currently, measuring serum cardiac troponin levels are highly recommended as a diagnostic biomarker [4,5,6,7,8]. However, in patients with certain conditions such as renal impairment, the concentrations of high sensitivity cardiac troponin are frequently elevated, which makes interpretation challenging [9,10,11]. Furthermore, measuring serum cardiac enzyme required invasive procedure. Therefore, it is important and beneficial to have a biomarker which is non-invasive and not affected by renal failure.

Titin is a component of sarcomeres, connecting Z and M lines, and is located in skeletal and cardiac muscle [12,13]. Previous studies have shown that the level of titin in muscle or serum is associated with neuromuscular disease or cardiac disease such as dilated cardiomyopathy (DCM) or AMI [14,15,16,17,18]. Vassiliadis et al. elucidated that titin-12670 fragment is present in both individuals with undiagnosed and diagnosed cardiovascular disease. In addition, titin levels were statistically significantly elevated in the AMI group, suggesting that titin is a useful serological marker in AMI [19]. However, its measurement in clinical practice was infeasible [20]. Recently, the breakdown product of titin, the N-terminal fragment, has become measurable in urine [21]. Several studies have demonstrated that urinary N-fragment titin is associated with Duchenne muscular dystrophy (DMD) and amyotrophic lateral sclerosis (ALS) in human subjects [22,23,24]. Moreover, it is reported that urinary N-fragment titin can be a biomarker to evaluate disease severity and prognosis in patients with ALS [24]. Nevertheless, no study has shown an association between urinary titin and AMI. We aimed to demonstrate that urinary titin N-fragment may be a non-invasive diagnostic and prognostic biomarker in AMI.

## 2. Results

### 2.1. Patient Characteristics

Figure 1 shows a selection of eligible patients and the enrolment process. One hundred and fifty-three patients were suspected of AMI, among whom 83 were included for the analysis. Suspected AMI was defined as an episode of anterior chest pain lasting more than 30 min and considered by the emergency physician or cardiologist to require catheterization. Arrhythmias and old myocardial infarction (OMI) were excluded from cases of suspected AMI. OMI was excluded based on ECG diagnosis.

Of the 70 patients who met the exclusion criteria, 58 patients did not have urinary tests, and 7 patients were receiving dialysis. Among the 83 included patients, 51 were diagnosed as AMI. The baseline characteristic of the study group is presented in Table 1. The mean age of the included patients was 73 ± 12.1 years, and 60 patients were male (72.3%). For comorbidities, cardiovascular disease in the AMI group was less frequent than in the non-AMI group (12.2% vs. 56.3%, *p* < 0.001). Apart from this, the differences of other comorbidities between these two groups were not statistically significant. Overall, 59% of patients were admitted to the emergency department within 3 h after symptom onset. The number of patients with chest pain was not statistically significant between the non-AMI and AMI groups (56.3% vs. 74.5%; *p* = 0.084). Nevertheless, ST-segment elevation was observed much more in the AMI group than the non-AMI group (88.2% vs. 28.1%; *p* < 0.001). Among blood exams on admission, cardiac enzymes, such as serum high-sensitivity troponin I (hsTnI), creatine kinase (CK), and creatine kinase-MB (CK-MB), were significantly higher in the AMI group compared to the non-AMI group (254 pg/mL vs. 47 pg/mL; *p* = 0.003, 165 U/L vs. 95 U/L; *p* = 0.003, 12.8 U/L vs. 6.0 U/L; *p* = 0.007, respectively).

### 2.2. Urinary Titin Level between AMI and Non-AMI Patients

Urinary titin levels on admission are shown in Figure 2. Among the patients admitted to the emergency department (ED) within 3 h after symptom onset, their urinary titin levels were not statistically different between the AMI and non-AMI group (4.47 [IQR 2.87–11.30] pmol/mg Cr (creatinine) vs. 4.37 [IQR 3.43–13.72] pmol/mgCr; *p* = 0.978). However, among the patients whose admission time was longer than 3 h after onset, the urinary titin level of the AMI group was significantly elevated compared to the non-AMI group (12.76 [IQR 5.87–16.68] pmol/mgCr vs. 5.13 [IQR 3.93–11.25] pmol/mgCr, *p* = 0.045). Moreover, among the patients in the AMI group, the patient group who were admitted longer than 3 h after onset showed significantly higher levels of urinary titin, compared to the group whose admission time was less than 3 h after symptom onset (12.76 [IQR 5.87–16.68] pmol/mgCr vs. 4.47 [IQR 2.87–11.30] pmol/mgCr, *p* = 0.008). Figure 3 shows the relationship between the level of urinary titin and time after symptom onset, as well as between CK-MB and time after symptom onset. The level of urinary titin was proportionally increased time-dependently, and the same pattern was seen with CK-MB. Table 2 shows the association between urinary titin and each parameter in all patients. Urinary titin was not associated with the sex, age, BMI, or eGFR. Moreover, eGFR was not associated with urinary titin in multiple linear regression analysis (Table 3). In contrast, urinary titin was significantly correlated with CK, CK-MB, and hsTnI (rs 0.586 *p* < 0.001, rs 0.280 *p* = 0.010 and rs 0.402 *p* < 0.001, respectively). Additionally, the AMI patient group showed a statistically significant association between urinary titin level and CK-MB, and hsTnI, while the non-AMI patient group did not show this association between them (AMI patient group: rs 0.314 *p* = 0.025, and rs 0.525 *p* < 0.001; non-AMI patient group: rs 0.213 *p* = 0.243, and rs 0.201 *p* = 0.287, respectively) (Appendix A).

The AMI group, which was admitted longer than 3 h, has significantly higher level of urinary titin compared to equivalent admission time in non-AMI group (12.76 [5.87–16.68] pmol/mgCr vs. 5.13 [3.93–11.25] pmol/mgCr, *p* = 0.045, respectively).

### 2.3. Level of Urinary Titin and Prognosis

Figure 4 shows the association between urinary titin level on admission and prognosis in all patients who are suspected of AMI. The urinary titin level of patients who were discharged, transferred to other facilities, and died during hospitalization were 4.90 [IQR 3.55–11.95] pmol/mgCr, 6.35 [IQR 4.02–12.57] pmol/mgCr, and 15.90 [IQR 13.46–22.61] pmol/mgCr, *p* = 0.023, respectively. Among these three groups, the difference in urinary titin levels between the group who died during hospitalization and who were discharged was statistically significant (*p* = 0.023). On the other hand, there was no statistical difference between the patients discharged and those transferred to other facilities (*p* = 0.768). In contrast, the transfer time from first symptom onset to hospital admission was not statistically different among each patient group (discharged: 120 [74–366] min; other facilities: 199 [115–786] min; and deceased: 194 [54–1200] min; home vs. transfer to other facilities, *p* = 0.416; home vs. deceased, *p* = 0.993; transfer to other facilities vs. deceased, *p* = 0.857).

## 3. Discussion

To our knowledge, this is the first study to evaluate urinary titin levels in patients with AMI. Our study brings several new insights into the diagnosis and prognosis of AMI. We found that N-fragment titin is excreted in urine in a time-dependent manner after the onset of symptoms in AMI patients, which has the advantage of being measurable many times without invasion. As a result, among patients admitted to the hospital longer than 3 h after symptom onset, the elevation of urinary titin levels in that AMI group was statistically significant compared to the non-AMI group. We also found that the level of urinary titin on admission in patients suspected of AMI was associated with their prognosis. Our results suggest that urinary titin can be a non-invasive biomarker to diagnose AMI and predict the prognosis.

Recently, there has been a focused discussion on the role of titin. Previous studies have shown that serum titin increases early on after symptom onset in patients with AMI [15,16]. These results are compatible with our results, which show that the titin N-fragment in AMI patients was excreted into urine early on after symptom onset. Moreover, J. Bogomolvas et al. showed that serum titin concentration was constant at symptom onset in AMI patients [16]. In contrast, in our study, urinary titin was proportionally increased time-dependently. It might be because serum titin is gradually excreted into urine. Since urinary titin increases in a time-dependent manner, the elevation of urinary titin in AMI patients transported to the ED after 3 h from symptom onset was statistically significant compared to non-AMI patients. This indicates that urinary titin could be useful to diagnose AMI as early as 3 h or longer after symptom onset. It is known that a significant increase in urinary titin is detected because of muscle damage due to exercise or muscle atrophy [25,26,27]. However, those study showed that fragmented titin appeared in urine 24 h or later after exercise or intensive care unit (ICU) admission. In contrast, a previous study demonstrated that the metalloproteinase-2 (MMP-2) matrix degrades titin after 10 min of reperfusion in a rat myocardial ischemia/reperfusion injury model [14]. Moreover, serum titin N2B exon, which is a cardiac-specific titin, was elevated incredibly early on after the onset of symptoms [16]. Kotter has previously shown the relationship between titin and ventricular remodeling in animals. They concluded that titin-based cardiac myocyte stiffening acutely after MI is partly mediated by interleukine-6 and is an important mechanism of remote myocardium to adapt to the increased mechanical demands after myocardial injury [28]. Since there was a significant increase in N-fragment titin in urine as early as 3 h after symptom onset in our cohort, the detected urinary titin in our study might be derived from cardiac muscle, instead of skeletal muscle, although the measured N-fragment urinary titin is not a cardiac-specific titin. In fact, age and sex which can influence muscle mass [29,30,31] were not correlated with urinary titin in our cohort. Nevertheless, cardiac enzymes, hsTnI, CK, and CK-MB, were associated with urinary titin. In addition, urinary titin was correlated with CK-MB and hsTnI in the AMI patient group despite no correlations being shown between them in the non-AMI patient group. Those results enhance the possibility that urinary titin in this study derives from cardiac muscle and reflects acute myocardial infarction. Another important finding is that eGFR did not show correlation with urinary titin in this cohort. It could indicate that measured urinary titin is not affected by renal function as there was no association between renal dysfunction (low eGFR) and high titin levels, which can be incredibly useful for the diagnosis of AMI since high-sensitivity cardiac troponin can be elevated in patients with renal dysfunction [11].

It is important to evaluate the patients’ prognosis for the management of AMI. CK-MB, which is an isoenzyme of CK and located mainly in the myocardium, is well known for its elevation reflecting myocardial cellular injury and predict infarct size in AMI [32]. Also, it is associated with mortality after percutaneous coronary intervention (PCI) [33,34,35,36,37,38]. In our study, urinary N-fragment titin increased proportionally, and the same manner was seen with CK-MB after symptom onset. Nevertheless, Figure 3 shows that CK-MB and urinary titin increases are temporally correlated. These time lags may be due to differences between blood and urine. Those result led us to investigate the relationship between urinary titin levels and mortality in patients suspected of AMI. Surprisingly, patients who died during hospitalization had higher urinary titin levels on admission than those who were discharged, and the difference was statistically significant, although the transfer time to the hospital after first symptom onset was not statistically different between those groups. Moreover, it is highly notable that urinary titin levels in the patients who died during hospitalization were also incredibly high compared to those who were transferred to other facilities, even though the difference is not statistically significant. Moreover, there is no statistical difference in the transfer time between them. It indicates that urinary titin can be a prognostic biomarker in patients who are suspected of AMI. To clarify this association between titin and deaths, we need to consider larger cases or ejection fraction (EF) data. AMI is one of the typical diseases presenting with HFrEF. EF is obtained by echocardiography in usual practice, but not for the purpose of this study, and we do not have sufficient data to examine its correlation with titin. Although further investigations are warranted, it could be one of many options to measure urinary titin when patients are admitted to the hospital to evaluate prognosis. 

This study has several limitations. This study included a selected patient population. In practice within facilities, urinary tests were not often conducted on patients diagnosed with non-AMI. Because we used urine samples taken for clinical purposes, we could not include patients who did not have urine test to avoid the delay of treatment. The follow-up term was relatively short. Moreover, although urinary titin is elevated independently of with renal function, urinary titin did not show an increase within 3 h of onset, whereas CK-MB revealed a significant increase from the hyperacute stage. We also measured N-fragment titin, which is not cardiac-specific. In addition, Myoglobin, another biomarker of myocardial necrosis, was not assessed and would have added valuable information. High sensitivity C-reactive protein, a biomarker of atherosclerosis, associated with arterial stiffness, can also predict acute cardiovascular events but it was also not assessed in our study [39].

In AMI patients, N-fragment titin appeared in urine immediately after symptom onset and show a significant difference as early as 3 h after symptom onset compared to non-AMI patients. Moreover, urinary titin level on admission might be associated with prognosis. Although urinary titin is a less predictive marker than CK-MB, it can be measured very easily and without invasion, and if it can be detected by filter paper discs instead of ELISA, we expect it to be clinically applicable. Therefore, urinary N-fragment titin could be a novel non-invasive diagnostic biomarker of AMI.

## 4. Materials and Methods

### 4.1. Study Design

We conducted a two-center, prospective observational study at Tokushima University Hospital and Tokushima Prefectural Central Hospital from January 2021 to November 2021. This study was approved at both clinical research ethics committees of Tokushima University Hospital (approval number 2593) and Tokushima Prefectural Central Hospital (approval number 1739). 

This study was registered in a clinical trial (University hospital Medical Information Network-Clinical Trials Registry: 000043419).

### 4.2. Patient Selection Criteria

We collected patients with suspected AMI at the emergency department (ED) consecutively during the study time. We excluded patients who did not have urine tests at the emergency department and who had been symptomatic for more than 72 h at arrival. We also exclude patients whose urinary titin N-fragment can be elevated by other causes: (1) surgery including coronary artery bypass grafting, (2) seizure, (3) trauma, (4) dialysis, or (5) primary neuromuscular disorder. 

### 4.3. Urinary Titin

Urinary titin was measured using an ELISA kit (27900 Titin N-Fragment Assay Kit, Immuno-Biological Laboratories Co. Ltd., Gunma, Japan), which measures the N-terminal fragment of urinary titin [20]. Urinary titin is creatinine (Cr) corrected to adjust for various physiological conditions, including renal function. Urinary titin was measured on admission. 

### 4.4. Troponin I, Creatine-Kinase, and Creatine Kinase-MB

Serum high-sensitivity troponin I (hsTnI), creatine-kinase (CK), and creatine kinase-MB (CK-MB) were measured as a routine clinical procedure at the emergency department. The following measurement tools were used: Troponin I (Fuji Rebio Co., Tokyo, Japan), CK (MERCK-1-TEST CK, Kanto Kagaku, Tokyo, Japan), and CK-MB (MERCK-1-TEST CK-MB, Kanto Kagaku, Tokyo, Japan, or the L-type Wako CK-MB mass, Wako Co., Kyoto, Japan). 

### 4.5. Acute Myocardial Infarction

AMI was diagnosed by coronary angiography. On the other hand, non-AMI was defined by no coronary infarction diagnosed by coronary angiography or diagnosis by two physicians (board-certified cardiologists or emergency physician). Cardiologists decided the necessity of coronary angiography and percutaneous coronary intervention. ST-segment change was also assessed by the two physicians.

### 4.6. Clinical Outcomes

All patients were followed up with during hospitalization. The primary outcome was the diagnostic utility of urinary titin for AMI. This diagnostic utility was further analyzed at different times after the first symptom onset: ≤3 h and >3 h. Secondary outcomes included the prognostic value of AMI.

### 4.7. Statistical Analysis

Continuous and normally distributed data were presented as mean values with standard deviation (SD), while non-normally distributed data were presented as median values with interquartile ranges (IQR). Normally distributed data were compared with Student’s *t*-test, while non-normally distributed data were compared with the Wilcoxon rank sum test. Categorical data were presented as number (%) and compared using the χ^2^ test. Correlations between two continuous variables were evaluated by Spearman’s rank coefficient correlation. A comparison of three or more groups was used for the Steel–Dwass test. The association between urinary titin and external variables were evaluated by multiple linear regression. In multiple regression analysis, urinary titin, CK, and BNP were log-transformed, and data analyses were conducted using JMP version 13.1.0 (SAS Institute Inc., Cary, NC, USA). All statistical tests were two-tailed, and a *p*-value < 0.05 was considered statistically significant.

## Figures and Tables

**Figure 1 ijms-25-00573-f001:**
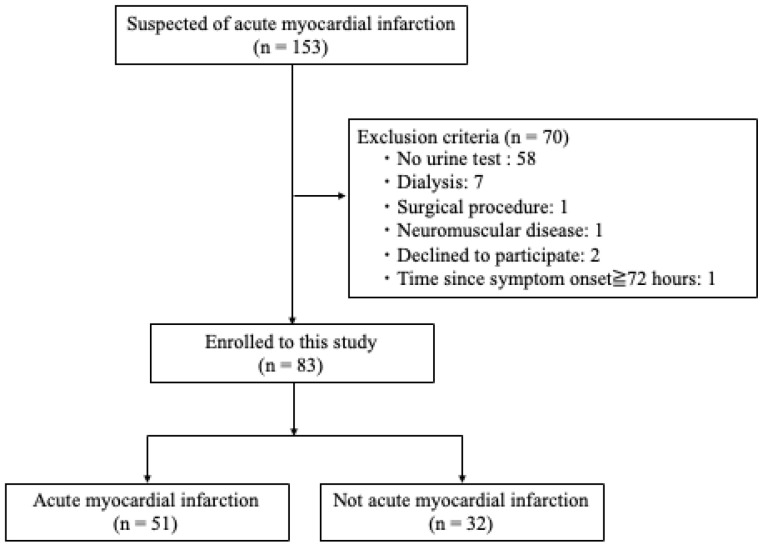
Patient selection.

**Figure 2 ijms-25-00573-f002:**
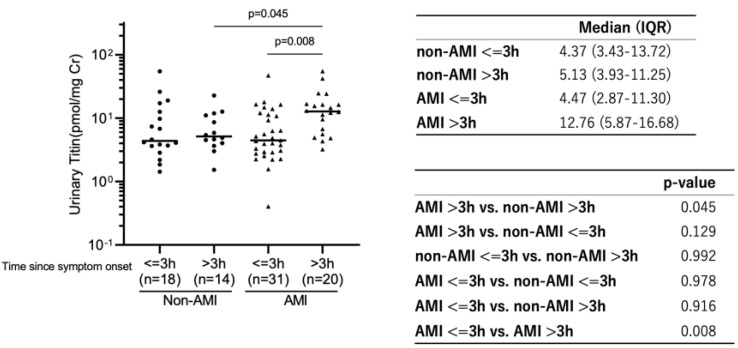
The difference of urinary titin levels based on different admission time in AMI and non-AMI patients.

**Figure 3 ijms-25-00573-f003:**
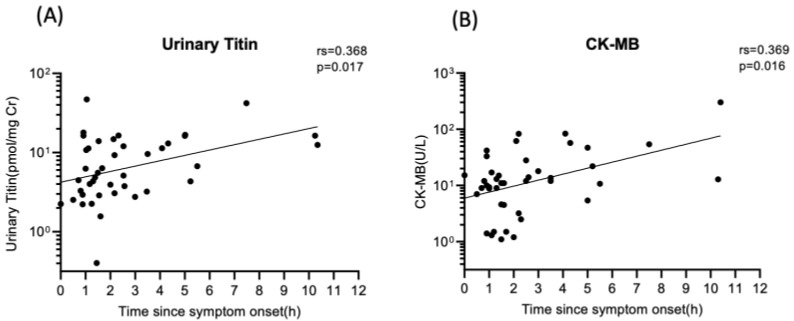
Correlation between time since symptom onset and urinary titin, and CK-MB. (**A**) The N-terminal fragment of titin was excreted into urine time-dependently after first symptom onset (rs 0.368, *p* = 0.017). (**B**) Serum CK-MB increases proportionally in time-dependent manner after symptom onset (rs 0.369, *p* = 0.016).

**Figure 4 ijms-25-00573-f004:**
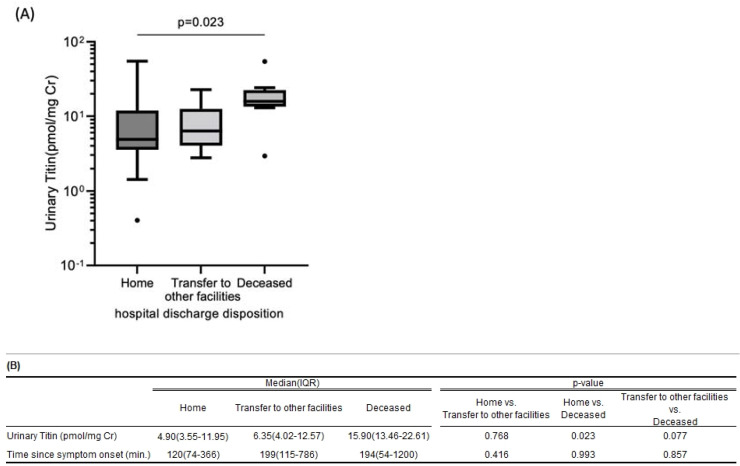
(**A**) Association between urinary titin level on admission and hospital discharge disposition. (**B**) Urinary titin levels in the patients who died during the hospitalization were significantly higher than in those who were discharged to home or transferred to other facilities. On the other hand, the difference in transferred time to hospital from first symptom onset was not statistically significant among each patient group.

**Table 1 ijms-25-00573-t001:** Patient characteristics.

				Median (IQR) unless Otherwise Noted
	All	Non-AMI	AMI	*p*-Value
	(n = 83)	(n = 32)	(n = 51)	(Non-AMI vs. AMI)
Age (years), mean ± SD	73.0 ± 12.1	76.1 ± 12.4	71.0 ± 11.6	0.063
Male sex, n (%)	60 (72.3)	22 (68.8)	38 (74.5)	0.568
BMI (kg/m^2^)	22.8 (20.8–26.2)	22.2 (20.2–26.0)	23.6 (21.2–26.5)	0.128
Comorbidities, n (%)				
Cardiovascular diseases	24 (29.6)	18 (56.3)	6 (12.2)	<0.001
Hypertension	54 (66.7)	21 (65.6)	33 (67.3)	0.872
Diabetes mellitus	35 (43.2)	10 (31.3)	25 (51.0)	0.079
Dyslipidemia	29 (35.8)	12 (37.5)	17 (34.7)	0.797
Renal failure	11 (13.6)	6 (18.8)	5 (10.2)	0.272
Former or current Smoker, n (%)	40 (48.8)	12 (37.5)	28 (56.0)	0.102
ECG: STEMI, n (%)	54 (65.1)	9 (28.1)	45 (88.2)	<0.001
Chest pain, n (%)	56 (67.5)	18 (56.3)	38 (74.5)	0.084
Hospital discharge disposition, n (%)				
Home	66 (79.5)	27 (84.4)	39 (76.5)	
Transfer to other facilities	9 (10.8)	4 (12.5)	5 (9.8)	
Deceased	8 (9.6)	1 (3.1)	7 (13.7)	
Time since symptom onset, n (%)				0.683
>3 h	34 (41.0)	14 (43.8)	20 (39.2)	
≤3 h	49 (59.0)	18 (56.3)	31 (60.8)	
Heart rate (beats/min)	73.0 (60.8–88.0)	73.0 (62.0–87.0)	73.0 (60.0–92.0)	0.789
Systolic blood pressure (mmHg), mean ± SD	138.1 ± 31.3	153.1 ± 27.8	129.4 ± 30.2	<0.001
Diastolic blood pressure (mmHg), mean ± SD	82.1 ± 21.4	86.5 ± 20.3	79.5 ± 21.8	0.158
SpO_2_ (%)	98.0 (95.8–99.0)	98.0 (97.0–99.0)	97.0 (95.0–99.0)	0.174
BNP (pg/mL)	93.5 (22.3–235.5)	152.7 (32.7–330.4)	56.5 (19.8–127)	0.027
BUN (mg/dL)	17.8 (13.9–21.1)	18.8 (13.5–20.8)	17.5 (14.7–22.0)	0.925
eGFR (mL/min/1.73 m^2^), mean ± SD	62.3 ± 21.9	63.6 ± 22.5	61.5 ± 21.7	0.684
Serum Cre (mg/dL)	0.9 (0.7–1.1)	0.8 (0.6–1.1)	0.9 (0.7–1.2)	0.250
Serum Troponin I (pg/mL)	156 (24–2,865)	47 (7–245)	254 (42–12,233)	0.003
Serum CK (U/L)	126 (81–306)	95 (68–201)	165 (98–520)	0.003
Serum CK-MB (U/L)	11.0 (4.1–22.0)	6.0 (1.5–15.0)	12.8 (7.0–33.0)	0.007
Urinary Titin (pmol/mg Cr)	5.58 (3.70–13.03)	4.75 (3.67–11.70)	6.35 (3.77–14.76)	0.335

BMI, body mass index; STEMI, ST-elevation myocardial infarction; SpO_2_, saturation of peripheral oxygen; BNP, brain natriuretic peptide; BUN, blood urea nitrogen; eGFR, estimated glomerular filtration rate; Cre, creatinine; CK, creatine kinase; CK-MB, creatine kinase-MB.

**Table 2 ijms-25-00573-t002:** Spearman’s rank correlation between urinary titin and parameters in all patients.

Wilcoxon Rank Sum Test		Median (IQR)
	Urinary Titin (pmol/mgCr)	*p*-Value
Sex		0.531
Male	6.31 (3.87–12.94)	
Femele	4.91 (3.63–13.96)	
Spearman’s rank correlation		
	**rs**	** *p* ** **-value**
Age	0.138	0.213
BMI	0.116	0.298
Systolic blood pressure	−0.020	0.862
Diastolic blood pressure	−0.057	0.613
SpO_2_	−0.200	0.071
CK	0.586	<0.001
CK-MB	0.280	0.010
Serum Hs-Troponin I	0.402	<0.001
BUN	0.311	0.004
BNP	0.460	<0.001
eGFR	−0.201	0.068

BMI, body mass index; SpO_2_, saturation of peripheral oxygen; BUN, blood urea nitrogen; BNP, brain natriuretic peptide; eGFR, estimated glomerular filtration rate; Hs-Troponin I, high-sensitivity Troponin I; CK, creatine kinase; CK-MB, creatine kinase-MB.

**Table 3 ijms-25-00573-t003:** Multiple linear regression analysis of factors affecting urinary titin in all Patients.

	β	*p*-Value
Age	0.019	0.878
CK (log)	0.465	<0.001
BNP (log)	0.298	0.014
eGFR	−0.074	0.502

CK, creatine kinase; BNP, brain natriuretic peptide; eGFR, estimated glomerular filtration rate.

## Data Availability

Data are available upon reasonable request for academic, non-commercial research purposes.

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
