# Peer review of "The Utility of Urinary Titin to Diagnose and Predict the Prognosis of Acute Myocardial Infarction"

_ijms, 2024, doi:10.3390/ijms25010573_

Round 1
Reviewer 1 Report
Comments and Suggestions for Authors
The present paper hypothesized that urinary N-fragment titin may be a biomarker for the diagnosis and prognosis in acute myocardial infarction.
Strengths: one of the first studies to evaluate urinary titin levels in patients with AMI.
A few changes are needed, as follows:
Please explain every abbreviation before using it!
Introduction, lines 50-51: You state: “We aimed that urinary titin N-fragment may be a non-invasive diagnostic and prognostic biomarker in AMI.” Please rephrase! Maybe: “We aimed to demonstrate that urinary titin N-fragment may be a non-invasive diagnostic and prognostic biomarker in AMI.”
Results, lines 72-75: Please rephrase! If you just replace “statistically significant” with “statistically different”, it would be easier to understand.
Discussion: Please add a few words about titin and ventricular remodeling (Titin-Based Cardiac Myocyte Stiffening Contributes to Early Adaptive Ventricular Remodeling After Myocardial Infarction. Circ Res. 2016 Oct 14;119(9):1017-1029. doi: 10.1161/CIRCRESAHA.116.309685. Epub 2016 Sep 20).
Please add a few words about the study: Clinical evaluation of a matrix metalloproteinase-12 cleaved fragment of titin as a cardiovascular serological biomarker. J Transl Med. 2012 Jul 6;10:140. doi: 10.1186/1479-5876-10-140
Study limitations: Myoglobin, another biomarker of myocardial necrosis was not assessed and would have added valuable information. High sensitivity C-reactive protein, a biomarker of atherosclerosis, associated with arterial stiffness, can also predict acute cardiovascular events (Links between High-Sensitivity C-Reactive Protein and Pulse Wave Analysis in Middle-Aged Patients with Hypertension and High Normal Blood Pressure. Dis Markers. 2019 Jul 17;2019:2568069. doi: 10.1155/2019/2568069), and was not assessed in the present study.
Materials and Methods, Clinical Outcomes: which is the time of follow up of your patients?
Author Response
Thank you for reviewing our manuscript entitled “The utility of urinary titin to diagnose and predict the prognosis of acute myocardial infarction”. We appreciate your valuable and helpful comments.
Comment; The present paper hypothesized that urinary N-fragment titin may be a biomarker for the diagnosis and prognosis in acute myocardial infarction. Strengths: one of the first studies to evaluate urinary titin levels in patients with AMI. A few changes are needed, as follows:
Comment#1 Please explain every abbreviation before using it!
Response
Thank you so much for your pointing out. We have added explanation for all abbreviation such as Cr and ED.
Comment#2
Introduction, lines 50-51: You state: “We aimed that urinary titin N-fragment may be a non-invasive diagnostic and prognostic biomarker in AMI.” Please rephrase! Maybe: “We aimed to demonstrate that urinary titin N-fragment may be a non-invasive diagnostic and prognostic biomarker in AMI.”
Response
We appreciated your suggestion and rephased as follows; We aimed to demonstrate that urinary titin N-fragment may be a non-invasive diagnostic and prognostic biomarker in AMI.” (L54-55)
Comment#3
Results, lines 72-75: Please rephrase! If you just replace “statistically significant” with “statistically different”, it would be easier to understand.
Response.
We have rephased following your suggestion. (L. 83)
Comment#4
Discussion: Please add a few words about titin and ventricular remodeling (Titin-Based Cardiac Myocyte Stiffening Contributes to Early Adaptive Ventricular Remodeling After Myocardial Infarction. Circ Res. 2016 Oct 14;119(9):1017-1029. doi: 10.1161/CIRCRESAHA.116.309685. Epub 2016 Sep 20).
Response.
Thank you very much for your comments. We add about titin and ventricular remodeling as; Kotter has previously shown the relationship between titin and ventricular remodeling in animals. they concluded that titin-based cardiac myocyte stiffening acutely after MI is partly mediated by interleukine-6 and is an important mechanism of remote myocardium to adapt to the increased mechanical demands after myocardial injury. (L. 218-222)
Comment#5
Please add a few words about the study: Clinical evaluation of a matrix metalloproteinase-12 cleaved fragment of titin as a cardiovascular serological biomarker. J Transl Med. 2012 Jul 6;10:140. doi: 10.1186/1479-5876-10-140
Review
Thank you so much for your comment. We added about the study as ; Vassiliadis et al. elucidated that titin-12670 fragment is present in both individuals with undiagnosed and diagnosed cardiovascular disease. In addition, titin levels were statistically significantly elevated in the AMI group, suggesting that titin is a useful serological marker in AMI. [19]. (introduction L. 45-48)
Comment#6
Study limitations: Myoglobin, another biomarker of myocardial necrosis was not assessed and would have added valuable information. High sensitivity C-reactive protein, a biomarker of atherosclerosis, associated with arterial stiffness, can also predict acute cardiovascular events (Links between High-Sensitivity C-Reactive Protein and Pulse Wave Analysis in Middle-Aged Patients with Hypertension and High Normal Blood Pressure. Dis Markers. 2019 Jul 17;2019:2568069. doi: 10.1155/2019/2568069), and was not assessed in the present study.
Response.
I appreciate your pointing that out. We added the sentences about myoglobin as study limitation. (L. 264-267)
Comments#6
Materials and Methods, Clinical Outcomes: which is the time of follow up of your patients?
Response
Thank you for your pointing out. All patients were followed up during the hospitalization. (L.314)
Reviewer 2 Report
Comments and Suggestions for Authors
Reviewing the manuscript entitled, “The utility of urinary titin to diagnose and predict the prognosis of acute myocardial infarction” by Arase M et al., this focuses on usefulness of urinary titin as a marker for predicting myocardial infarction prognosis. Although this is an interesting manuscript that shows prospective observational study, there are some inconsistencies and questions. The authors need to respond to the following concerns.
In patient eligibility of Figure 1, what does “Suspected AMI” mean? The authors should provide more specific findings.
In Figure 1, does the exclusions criteria include OMI and arrhythmia? In Figure 4, the author analyzes the prognosis prediction of urinary titin by discharge, transfer, and death. If the subject has OMI or arrhythmia, it will affect the results as a confounding factor.
The authors mentioned that urinary titin is independent of renal function. On that basis, why did urinary titin not show an increase within 3 hours of onset? CPK-MB revealed a significant increase from the hyperacute stage. Nevertheless, Figure 3 showed that CPK-MB and urinary titin increases are temporally correlated. It seems contradictory. The authors need to provide it.
In Figure 4, was the cause of death due to heart failure due to present AMI? There are only a few cases, and the categories of discharge, transfer, and death as predictors of prognosis for urinary titin are too arbitrary. AMI is one of the typical diseases presenting with HFrEF. For example, can you show correlation with EF determined by echocardiography?
Do the authors have serum titin data?
At least, the results indicate that urinary Titin does not appear to be a more predictive marker than CPK-MB. The authors need to describe advantages of urinary titin as a predictive marker for AMI.
From line 165 to 166, the authors mentioned “We found that N-fragment titin is excreted into urine time-dependently at very early time 165 from the first symptom onset in AMI patients.” This is an expression that does not match the results. I don't think it's useful for early diagnosis. The authors need to modify it.
From line 217 to 219, the authors mentioned “Although 217 further investigation will be warranted, it could be very useful to measure urinary titin 218 when patients are admitted at the hospital to evaluate prognosis.” Is "very useful" a bit of an exaggeration?
Comments on the Quality of English LanguageThere is no major problems in this manuscript.
Author Response
Comments#1
Reviewing the manuscript entitled, “The utility of urinary titin to diagnose and predict the prognosis of acute myocardial infarction” by Arase M et al., this focuses on usefulness of urinary titin as a marker for predicting myocardial infarction prognosis. Although this is an interesting manuscript that shows prospective observational study, there are some inconsistencies and questions. The authors need to respond to the following concerns.
In patient eligibility of Figure 1, what does “Suspected AMI” mean? The authors should provide more specific findings. In Figure 1, does the exclusions criteria include OMI and arrhythmia? In Figure 4, the author analyzes the prognosis prediction of urinary titin by discharge, transfer, and death. If the subject has OMI or arrhythmia, it will affect the results as a confounding factor.
Response
Thank you for reviewing our manuscript entitled “The utility of urinary titin to diagnose and predict the prognosis of acute myocardial infarction”. We appreciate your valuable and helpful comments.
As for first comment, we apologize that we did not clearly indicate our Criteria. In the Methods section, we have included the criteria for suspecting AMI and also how we excluded OMI and arrhythmias.
Suspected AMI was defined as an episode of anterior chest pain lasting more than 30 minutes and considered by the emergency physician or cardiologist to require catheterization. (L60-64)
Arrhythmias and old myocardial infacrction (OMI) were excluded from cases of suspected AMI. OMI was excluded based on ECG diagnosis. (L64-65)
Commnents#2
The authors mentioned that urinary titin is independent of renal function. On that basis, why did urinary titin not show an increase within 3 hours of onset? CPK-MB revealed a significant increase from the hyperacute stage. Nevertheless, Figure 3 showed that CPK-MB and urinary titin increases are temporally correlated. It seems contradictory. The authors need to provide it.
Response
In the present study, we assumed no renal dysfunction (low eGFR) because there was no association between renal dysfunction (low eGFR) and high titin levels. However, as you mention, it did not correlate with CK-MB, which is emerged in the hyperacute phase. We believe this is the difference between blood and urine. We added about this as limitation of urinary Titin. (L. 212-213)
Comment#3
In Figure 4, was the cause of death due to heart failure due to present AMI? There are only a few cases, and the categories of discharge, transfer, and death as predictors of prognosis for urinary titin are too arbitrary. AMI is one of the typical diseases presenting with HFrEF. For example, can you show correlation with EF determined by echocardiography?
Response
As you pointed out, we understand that the number of cases is small and that there are limitations in asserting urinary titin as a prognostic predictor, and we have reworded our statement. In addition, EF is obtained by echocardiography in the usual practice, but not for the purpose of this study, and we do not have sufficient data to examine the correlation with titin, so we have listed it as a limitation. (L. 252-257)
Comment#4
Do the authors have serum titin data?
At least, the results indicate that urinary Titin does not appear to be a more predictive marker than CPK-MB. The authors need to describe advantages of urinary titin as a predictive marker for AMI.
Response
Since there is no ELISA that can measure in serum, mass spectrometry is required, so we were not able to measure it at this time. We are currently developing serum ELISA and will try to report the results including the relationship with CPK-MK in the next issue. We agree that urinary titin is a less predictive marker than CPK-MB. We have added it to the limitation as noted above. However, on the other hand, urinary titin can be measured very easily and without invasion, and if it can be detected by filter paper discs instead of ELISA, we expect it to be clinically applicable.
Comment#5
From line 165 to 166, the authors mentioned “We found that N-fragment titin is excreted into urine time-dependently at very early time from the first symptom onset in AMI patients.” This is an expression that does not match the results. I don't think it's useful for early diagnosis. The authors need to modify it.
Response
Thank you very much for your pointing out. We modified our expression as ; We found that N-fragment titin is excreted in urine in a time-dependent manner after the onset of symptoms in AMI patients, which has the advantage that it can be measured many times without invasion. (L. 193-196)
Comment#6
From line 217 to 219, the authors mentioned “Although further investigation will be warranted, it could be very useful to measure urinary titin when patients are admitted at the hospital to evaluate prognosis.” Is "very useful" a bit of an exaggeration?
Response
We agree with your comment. We modified as; Although further investigation will be warranted, it could be one of option to measure urinary titin when patients are admitted at the hospital to evaluate prognosis. (L. 255-257)
Round 2
Reviewer 2 Report
Comments and Suggestions for Authors
This is an acceptable quality after minor English editing.
Comments on the Quality of English LanguageMinor English editing is required.